# JNK Signaling in Stem Cell Self-Renewal and Differentiation

**DOI:** 10.3390/ijms21072613

**Published:** 2020-04-09

**Authors:** Takashi Semba, Rachel Sammons, Xiaoping Wang, Xuemei Xie, Kevin N. Dalby, Naoto T. Ueno

**Affiliations:** 1Section of Translational Breast Cancer Research, Department of Breast Medical Oncology, The University of Texas MD Anderson Cancer Center, Houston, TX 77030, USA; TSemba1@mdanderson.org (T.S.); xiwang@mdanderson.org (X.W.); XXie2@mdanderson.org (X.X.); 2Morgan Welch Inflammatory Breast Cancer Research Program and Clinic, The University of Texas MD Anderson Cancer Center, Houston, TX 77030, USA; 3Division of Chemical Biology and Medicinal Chemistry, College of Pharmacy, The University of Texas at Austin, Austin, TX 78712, USA; rachel.m.sammons@utexas.edu (R.S.); dalby@austin.utexas.edu (K.N.D.); 4Department of Oncology, Dell Medical School, The University of Texas at Austin, Austin, TX 78712, USA

**Keywords:** JNK, C-JUN N-terminal kinase, stem cell, cancer stem cell, stem cell niche, tumor microenvironment, WNT, NOTCH1

## Abstract

C-JUN N-terminal kinases (JNKs), which belong to the mitogen-activated protein kinase (MAPK) family, are evolutionarily conserved kinases that mediate cell responses to various types of extracellular stress insults. They regulate physiological processes such as embryonic development and tissue regeneration, playing roles in cell proliferation and programmed cell death. JNK signaling is also involved in tumorigenesis and progression of several types of malignancies. Recent studies have shown that JNK signaling has crucial roles in regulating the traits of cancer stem cells (CSCs). Here we describe the functions of the JNK signaling pathway in self-renewal and differentiation, which are essential features of various types of stem cells, such as embryonic, induced pluripotent, and adult tissue-specific stem cells. We also review current knowledge of JNK signaling in CSCs and discuss its role in maintaining the CSC phenotype. A better understanding of JNK signaling as an essential regulator of stemness may provide a basis for the development of regenerative medicine and new therapeutic strategies against malignant tumors.

## 1. Introduction

C-JUN N-terminal kinases (JNKs) are members of the mitogen-activated protein kinase (MAPK) family. Various stimuli activate them, such as environmental stresses [1], inflammatory cytokines [2], and growth factors [3]. The activated JNKs catalyze the phosphorylation of many substrates, resulting in the alteration of gene expression programs and, ultimately, a variety of cellular signaling processes, such as cell proliferation [4], migration [5], and apoptosis [6]. Additionally, the JNKs play significant roles in programs such as embryonic development [7], neural functions [8], wound healing [9], immunity [10], metabolic diseases [11], and tumor progression [12].

There are three JNK genes—*JNK1*, *JNK2*, and *JNK3*—from which multiple isoforms are expressed [13]. While the JNK1 and JNK2 proteins are ubiquitous in human cells, JNK3 is found predominantly in the central nervous system [14]. Similarly to other MAPK family members, such as the ERKs and p38 MAPKs, the JNKs are activated through multi-tiered phosphorylation cascades (Figure 1) [15]. External stimuli activate the MAP3Ks, such as MEKK1 [16], TRAF [17], ASK1 [18], TAK1 [19], HPK1 [20], and MLK3 [21]. The MAP3Ks transduce signals to the MAP2Ks, MKK7, and MKK4, which synergistically activate the JNKs through phosphorylation of a Thr and Tyr, respectively [22]. Once activated, the JNKs phosphorylate a variety of downstream substrates, including the transcription factors C-JUN [23], JUND [24], ATF2 [25], and ELK1 [26].

Over the past decade, evidence has emerged for the role of the JNK signaling pathway in certain types of stem cells. Stem cells can maintain themselves through continuous cycles of cell division (self-renewal) or the generation of different types of daughter cells (differentiation). Stem cells have indispensable roles in ontogenesis, embryonic development, and organ homeostasis [27]. Targeted disruption of JNK signaling-related genes such as *JNK1, JNK2, MKK4, MKK7,* and *JUN* causes disorders in embryonic development [28,29,30]. Furthermore, the activity of JNK signaling affects the proliferation and differentiation of tissue-specific stem cells, which mediate tissue homeostasis and regeneration [31,32,33]. As described below, recent evidence has identified several aspects of JNK signaling that regulate self-renewal and differentiation. Here we provide an overview of stem cells and describe the current understanding of the function of JNK signaling within various types of stem cells and between stem cells and their microenvironment.

## 2. Overview of Stem Cells

### 2.1. Normal Stem Cells

Normal stem cells are unspecialized quiescent cells found in embryonic, fetal, and adult tissues that replicate over long periods (self-renewal) until differentiating into more specialized cells. They exhibit an ability to transdifferentiate and dedifferentiate, as well as tolerance to toxic insults [34]. They may be distinguished from terminally differentiated somatic cells by the expression patterns of cell surface markers, signaling pathway-related intracellular markers, transcription factors, and enzymatic markers [35]. 

Normal stem cells are classified into totipotent, pluripotent, and multipotent cells (Figure 2A). Totipotent stem cells exhibit an ability to self-renew through asymmetric cell division and have the capability of differentiating into all cell types found in the body. Thus, they can develop into the three primary germ cell layers of the early embryo: the endoderm, the mesoderm, and the ectoderm. Additionally, they can differentiate into extra-embryonic tissues such as the placenta [36]. The only known totipotent cells are embryonic cells within the first couple of cell divisions following fertilization. 

Pluripotent stem cells, like totipotent stem cells, also possess the ability to differentiate into cells of all three germ cell layers of the early embryo. There are two types of pluripotent stem cells, including embryonic stem cells (ESCs) and induced pluripotent stem cells (iPSCs). ESCs, isolated from the inner cell mass of blastocysts [37,38], are considered pluripotent and can differentiate into islet cells [39], hepatocytes [40], neural precursors [41], endothelial cells [42], cardiomyocytes [43], and hematopoietic cells [44]. However, unlike totipotent stem cells, pluripotent stem cells cannot differentiate into extra-embryonic tissue. iPSCs were established by Yamanaka’s group [45] by transducing four transcription factors (Oct4, Sox2, c-Myc, and Klf4) into murine fibroblasts. Like ESCs, iPSCs also have the potential to differentiate into various types of cells, including retina [46], liver [47], pancreatic islets [48], brain [49], and blood vessels [50]. 

Multipotent stem cells have more limited differentiation potential but still can give rise to various types of lineage-specific cells. Adult tissue-specific stem cells are considered multipotent. These include hematopoietic stem cells (HSCs), intestinal stem cells (ISCs), and neural stem cells (NSCs). These cells are present in the respective tissues, where they maintain tissue regeneration and participate in wound repair [51]. Multipotent mesenchymal stromal cells (MSCs) are the fibroblast-like plastic-adherent cells derived from bone marrow and other tissues, which contain a subpopulation of stem cells of mesenchymal lineages [52]. According to the criteria proposed by the International Society for Cellular Therapy, cultured MSCs must express CD105, CD73 and CD90, and lack expression of CD45, CD34, CD14 or CD11b, CD79a, or CD19 and HLA-DR surface molecules and possess the ability to differentiate into osteocytes, chondrocytes, and adipocytes in vitro [53]. It should be noted that current standard culture conditions for MSCs do not isolate homogenous stem cell populations [54]. 

### 2.2. Cancer Stem Cells

Cancer stem cells (CSCs) are cancer cells with stem cell-like properties. The CSC model postulates that tumor cells are heterogeneous and composed of a hierarchical population of cells with CSCs at the top. The CSCs exhibit sole tumorigenic potential, with the bulk of the tumor consisting of proliferative non-tumorigenic cells derived from CSCs [55] (Figure 2B). Dick et al. [56] first identified CSCs based on cell surface markers as a result of the fractionation of human acute myeloid leukemia (AML) cells. Since normal HSCs show a CD34^+^CD38^−^ cell-surface phenotype, they purified the CD34^+^CD38^−^ subpopulation from primary human AML cells using fluorescence-activated cell sorting (FACS). They then demonstrated that leukemic engraftment in immunodeficient mice exclusively initiated from this subpopulation [56]. Applying these concepts and experimental approaches, such as fractionation of tumor cells based on cell-surface markers shared with normal stem cells, lineage tracing [57], and in vivo limiting dilution assays to determine tumor-initiation ability, has resulted in mounting evidence for the existence of CSCs, not only in hematopoietic malignancies but also solid tumors, including breast [58], brain [59], ovary [60], and colorectal [61] cancers. 

Both tumorigenesis and tumor progression may involve CSCs [55]. While conventional chemotherapy and radiotherapy eradicate non-CSC cancer cells, CSCs can survive such treatment because of cell cycle dormancy and enhanced protective mechanisms against oxidative stress [62]. CSCs also express high levels of drug efflux pumps such as ABC transporters [63] and a high level of activity of detoxifying enzymes such as aldehyde dehydrogenase 1 (ALDH1) [64], which impart therapeutic resistance to CSCs and can thus facilitate disease recurrence (Figure 2C). The epithelial–mesenchymal transition (EMT), which plays a crucial role in tumor invasion and metastasis, may also involve CSCs [65]. Thus, therapeutic strategies targeting CSCs represent promising approaches for the prevention of cancer metastasis and recurrence.

In this review, we will focus on the role of JNK signaling in (1) pluripotent stem cells, including ESCs and iPSCs; (2) multipotent adult tissue-specific stem cells, including HSCs, ISCs, and NSCs; and (3) CSCs. 

## 3. JNK Signaling in Pluripotent Stem Cells

### 3.1. Embryonic Stem Cells

Although there have been a limited number of studies related to the role of JNK signaling in maintaining the stemness or supporting the differentiation of ESCs, recently developed analytic approaches in various fields of molecular biology have begun to reveal roles for JNK signaling in the maintenance of ESC stemness. A study comparing gene expression between undifferentiated and differentiated human ESCs (hESCs) showed that some genes involved in the JNK signaling pathway, such as MAP4K1 (HPK1), MAP3K7 (TAK1), and JUN (C-JUN), were downregulated during differentiation [66], implying the potential contribution of JNK signaling to the maintenance of hESCs in the undifferentiated state. Brill et al. [67] conducted a phosphoproteomic analysis of hESCs using quantitative mass spectrometry and found that JNK signaling–related molecules such as TRAF4, MLK4, CRKL, and MINK1 are phosphorylated more often in undifferentiated hESCs than in hESC derivatives. Treating hESCs with JNK inhibitors (e.g., JNK inhibitor II (SP600125) [68] and JNK inhibitor III [69]) allowed further assessment of the role of JNK signaling in undifferentiated hESCs. Inhibitor treatment resulted in cellular differentiation and decreased OCT4 and NANOG expression in hESCs [67], suggesting that JNK signaling facilitates maintenance of the undifferentiated status of hESCs. However, SP600125 inhibits other kinases with similar potencies to its inhibition of JNK1, 2, and 3 [68,70]. Xu et al., in experiments utilizing Jnk1^−/−^, Jnk2^−/−^, and Jnk1^−/−^Jnk2^−/−^ murine ESCs (mESCs), demonstrated the effect of Jnk gene depletion on the stemness of mESCs [71]. For example, they showed that Jnk1^−/−^ and Jnk1^−/−^Jnk2^−/−^ mESCs, but not Jnk2^−/−^ mESCs, expressed Sox17 and Hnf1 genes, which are required for differentiation of definitive endoderm (DE) and visceral endoderm, respectively, and that these genes were expressed even before induction of differentiation. These results imply that JNK1 could have a suppressive role in the endodermal differentiation of mESCs. Interestingly, the proliferation rates of wild-type and Jnk1^−/−^ mESCs were similar, and Jnk2^−/−^ and Jnk1^−/−^ Jnk2^−/−^ mESCs proliferated even more rapidly than wild-type mESCs, indicating a possible role of JNK2 in inhibiting proliferation of mESCs [71].

Furthermore, recently Li et al. [72] conducted genome-scale CRISPR/Cas9 screening to discover the regulators of DE differentiation of hESCs and uncovered JNK/C-JUN pathway members as critical barriers to DE differentiation. Comprehensive knockout screens revealed that targeting core members of the JNK pathway, including MAP3K1 (MEKK4), MAP2K4 (MKK4), MAP2K7 (MKK7), MAPK8 (JNK1), and JUN (C-JUN), inhibited the differentiation of hESCs into SOX17^+^CXCR4^+^ DE cells. Using a JNK inhibitor, JNK-IN-8 [73], and epigenetic sequencing, they also demonstrated that C-JUN co-occupies transcriptional enhancers of ESC genes with OCT4, NANOG, and SMAD2/3 and maintains chromatin accessibility. Additionally, C-JUN impedes SMAD2/3 binding patterns that are associated with DE enhancers [72] (Figure 3A).

These findings suggest that the JNK signaling pathway has suppressive roles in differentiation and could maintain the undifferentiated status of ESCs. Given that evidence so far suggests non-redundant roles of each JNK isoform in differentiation and proliferation of ESCs, further studies to address the difference between JNK isoforms in the functions of lineage-specific differentiation and self-renewal in ESCs will be interesting. 

### 3.2. Induced Pluripotent Stem Cells

The role of JNK signaling in the stemness of iPSCs is less studied than in ESCs. However, the available evidence suggests that JNK signaling is involved in the reprogramming of somatic cells into iPSCs. More than a decade after the discovery of iPSCs, the efficacy of reprogramming of somatic cells has improved but remains low; barriers to reprogramming include transcription factors, microRNAs, and DNA methylation [74]. JNK1 and JNK2 phosphorylate KLF4, one of the “Yamanaka factors” needed to reprogram somatic cells and negatively regulate KLF4 activity in the reprogramming of MEFs [75] (Figure 3B). However, Neganova et al. [76] reported that JNK pathway members, including MKK4, MKK7, and JNK1/2, were activated during reprogramming of human dermal fibroblasts, and inhibition of JNK signaling activity by a JNK inhibitor or siRNA knockdown significantly reduced iPSC generation. More research is needed to understand these conflicting findings and the role of JNK signaling in maintaining the stemness of iPSCs. 

## 4. JNK Signaling in Adult Tissue-Specific Stem Cells

### 4.1. Hematopoietic Stem Cells 

HSCs are multipotent stem cells from which all types of hematopoietic cell lineages differentiate. HSCs predominantly reside in their niche, such as bone marrow, and mostly stay quiescent in the absence of stress. When stimulated by injury or infection, HSCs exit quiescence, and rapidly repopulate the entire hematopoietic system via multi-lineage differentiation [77]. C-FOS, one of the targets of JNK, has been shown to negatively control the entry of murine HSCs into the cell cycle and to maintain them in a dormant state [78]. Indeed, a recent study revealed that JNK inhibition by JNK-IN-8 enhanced the self-renewal of human HSCs through the downregulation of C-JUN phosphorylation [79] (Figure 3C). These results suggest that the JNK signaling pathway maintains HSCs in a quiescent state.

### 4.2. Intestinal Stem Cells

The human intestinal epithelium, which is composed of enterocytes, goblet cells, endocrine cells, and Paneth cells, is actively replaced every few days throughout life. ISCs reside at the bottom of the crypt of the intestine and can differentiate into every type of these functional epithelial cells to maintain intestinal epithelial cell turnover [80]. ISCs are known to express LGR5, which is a receptor for R-spondins, agonists of WNT signaling [81]. WNT signaling plays a critical role in maintaining stemness in ISCs [82]. However, recent studies have revealed that JNK signaling is also involved in regulating the stemness of ISCs. Using transgenic mice harboring constitutively active JNK1 under the Villin promoter, Sancho et al. [32] demonstrated that gut-specific JNK1 activation stimulated the cell cycle and differentiation of ISCs, resulting in increased intestinal cell proliferation [32]. Investigating underlying mechanisms, they found that the JNK/C-JUN pathway regulated WNT target genes and contributed to the upregulated expression of WNT pathway genes, including Ccdn1, Axin2, and Lgr5, in ISCs, as well as the proliferation of progenitor cells (Figure 3D). Such crosstalk of the JNK and WNT signaling pathways in ISCs is present in Drosophila, where it regulates the regenerative proliferation of ISCs and promotes their differentiation into progenitor cells in response to injury [83]. Thus, JNK signaling has positive roles in the proliferation and differentiation of ISCs, unlike in other types of stem cells, as described above. This divergence may suggest a cell type-dependent and/or JNK isoform-dependent function of the JNK signaling pathway, and further studies will be needed to address this question.

### 4.3. Neural Stem Cells

Adult NSCs locate to two sites of the brain: the subventricular zone of the lateral ventricle wall and the subgranular zone of the hippocampal dentate gyrus. They can generate multiple neural lineages, including neurons and astrocytes [84]. Bengoa-Vergniory et al. [85] showed that switching from WNT/β-catenin signaling to JNK signaling occurred during neuronal differentiation in human NSCs derived from ESCs and iPSCs. They demonstrated that a reduction in canonical WNT/β-catenin pathway activity and an increase of WNT/AP-1 non-canonical pathway activity accompanied neural differentiation. They also showed that inhibiting JNK signaling using JNK-IN-8 or gene silencing of ATF2 reduced WNT3A–mediated neuronal differentiation [85] (Figure 3E). JNK signaling also promotes differentiation of murine NSCs from mESCs by mediating phosphorylation of STAT1/3, which upregulate neuronal proteins including GAP-43, neurofilament, βIII-tubulin, and synaptophysin [86]. These results suggest positive roles of WNT-JNK or JNK-STAT signaling in NSC differentiation.

## 5. JNK Signaling in Cancer Stem Cells

JNK signaling mediates apoptosis [87], a process known to suppress tumorigenesis. Thus, it is not surprising that JNK signaling can play a negative role in cancer development. Indeed, loss-of-function mutations in the MAP2K4 gene (which encodes MKK4) exist in approximately 5% of tumors, and multiple lines of evidence indicate that the activity of MKK4 can suppress tumor progression [88]. Moreover, MKK7 is also known to function as a tumor suppressor. For example, MKK7 can promote senescence through p53 in response to oncogene activation and thereby negatively regulate tumorigenesis [89]. However, JNK signaling is also involved in cell survival and implicated in pro-tumorigenic function [14]. For instance, in a mutant Kras-driven lung cancer mouse model, conditional knockout of both Jnk1 and Jnk2 suppressed lung tumor formation, suggesting that JNK signaling plays a positive role in tumorigenicity [12]. There is also evidence that JNK signaling is involved in the protection of tumor cells from undergoing premature senescence by preventing mitochondrial reactive oxygen species production through activation of BCL-2 [90]. Thus, the role of JNK signaling in cancer development is likely to vary in a cell type and context-dependent manner. 

Accordingly, the association of the JNK signaling pathway with CSCs is also complicated. Some evidence suggests that JNK signaling does not affect CSC activity in specific types of cancer. In contrast, other evidence indicates that JNK signaling contributes to promoting aggressiveness for various kinds of malignancies through maintaining CSC properties such as self-renewal, drug resistance, and tumor-initiating ability. 

Girnius et al. [91] reported that JNK signaling does not regulate CSCs. They showed that in a Jnk-knockout breast cancer mouse model, Wap-Cre^+/−^:Trp53^LoxP/LoxP^:Jnk1^LoxP/LoxP^:Jnk2^LoxP/LoxP^ (JNK^KO^) mice exhibited more rapid breast tumor formation than control Wap-Cre^+/−^:Trp53^LoxP/LoxP^ (JNK^WT^) mice, implicating the role of JNK signaling as a tumor suppressor. They showed that tumor cells derived from JNK^KO^ mice exhibited no difference in proliferation, migration, or invasion phenotypes or in stem cell activity, such as sphere formation in vitro compared to tumor cells from JNK^WT^ mice. Moreover, no differences are apparent between the growth of JNK^WT^ and JNK^KO^ tumors in orthotopically transplanted mice. These findings implicate the suppressive role of JNK signaling in the initiation of murine breast cancer and the acceleration of tumor development, but only a minor role for JNK signaling in the growth of established tumors [91]. Ohta et al. [92] demonstrated that depletion of the histone H3K4 demethylase JARID1B by shRNA knockdown induced cellular senescence associated with JNK phosphorylation and reduced CSC population in colorectal cancer cells [92]. These results imply that JNK signaling has a suppressive role in CSC proliferation through the induction of premature senescence.

In contrast, we reported that the JNK/C-JUN signaling pathway promotes the CSC phenotype of triple-negative breast cancer (TNBC) through the upregulation of NOTCH1 [93]. Inhibiting JNK/C-JUN by siRNA-mediated knockdown or the JNK inhibitor JNK-IN-8 reduced CSC traits such as anchorage-independent growth and ALDH1 activity, as well as the migration and invasion activity of TNBC cells. Suppression of JNK signaling resulted in the downregulation of NOTCH1, which is a critical player in regulating self-renewal and cell fate determination of mammary stem cells (Figure 4). We also confirmed that C-JUN regulates NOTCH1 expression using a luciferase reporter assay and demonstrated that JNK-IN-8 repressed tumor growth in a TNBC xenograft mouse model via inhibition of CSC properties, including NOTCH1 and ALDH1 expression. These results support the role of JNK signaling in maintaining CSC properties and in crosstalk with NOTCH signaling [93]. 

Nasrazadani and Van Den Berg [94] also identified a role for JNK signaling in TNBC tumorigenesis that they attributed to the JNK2 isoform. In the metastatic 4T1.2 mammary tumor cell line, the shRNA knockdown of JNK2 reduced invasion by 80%. In vivo, shJNK2-expressing 4T1.2 cells in mice showed impaired tumor growth and reduced lung metastases. Jnk2^−/−^ mice also exhibited reduced bone metastasis, which was associated with impaired osteoclast differentiation resulting from JNK2 regulation of RANK and RANK ligand expression [94]. Further studies are needed to similarly evaluate the potential roles of JNK1.

A growing amount of evidence also supports a role for JNK signaling in maintaining glioma stem cells. Six glioblastoma cell lines expressed higher JNK activation in their stem cell populations than in their differentiated cell populations [95], and JNK activation correlates with glioma histological grade [96]. Matsuda et al. [95] found that JNK inactivation via siRNA or SP600125 reduced stemness in glioblastoma cells, as measured by spheroid formation and marker expression (nestin, SOX2, Musashi-1), and induced differentiation. Additionally, transient inactivation of JNK by either pharmacological or genetic methods in a mouse model of glioblastoma generated a sustained loss of tumor-initiating capacity. While some studies indicate that JNK1 and JNK2 both contribute to the CSC phenotype in gliomas [95,97], others have implicated JNK2 as the predominant acting isoform [98]. 

In pediatric patients with T-cell acute lymphoblastic leukemia (T-ALL), Shen et al. [99] showed that KLF4 expression was suppressed, which resulted in activation of MKK7 as well as the downstream JNK pathway, including JNK, ATF2, and C-JUN. They also showed that loss of Klf4 expanded NOTCH1-induced leukemia-initiating cells in mouse T-ALL models, and pharmacological inhibition of JNK using AS602801, JNK-IN-8, and CC401 reduced tumor progression of human T-ALL [99], implying potential collaboration between JNK and NOTCH signaling for strengthening tumor-initiation ability in CSCs.

There is evidence to support the JNK pathway regulating CSCs in numerous other cancers as well. For example, Okada et al. [100] used a combination of SP600125 treatment and siRNA knockdown of JNK1 and JNK2 to show that K-RAS/JNK signaling maintains stemness in pancreatic CSCs. JNK signaling also maintains self-renewal and tumor-initiation capabilities of ovarian CSCs. Seino et al. [101] found that in CSCs cultured from A2780 ovarian cancer cells, transient siRNA knockdown of JNK1 and JNK2 resulted in reduced expression of C-JUN and the stem cell markers NANOG, SOX2, and nestin. In a mouse intestinal model, Sancho et al. [32] found that transgenic expression of a constitutively active JNK1 fusion protein (JNKK2-JNK1^ΔG^) showed an increase in ISC proliferation and villus length. They attributed this result to a positive feedback loop that links JNK and WNT signaling, in which the WNT target Tcf4 is also a gene target of C-JUN, and C-Jun is a target of the TCF4/β-catenin complex. In the same study, JNKK2-JNK1^ΔG^ showed increased tumorigenesis in a model of colitis-induced intestinal carcinogenesis. Interestingly, this result was not observed in mice with Apc mutant–induced intestinal tumorigenesis, likely indicating that different signaling outcomes arise from different endogenous JNK pathway stimuli [32]. 

Further studies are needed to elucidate such paradoxical aspects of JNK signaling in cancer development and CSCs. Much of the evidence so far is based on experiments using pan-JNK inhibitors or compound disruption of JNK expression. The relative expression levels of the JNK proteins in each type of tumor cell is poorly understood. Each isoform of JNK likely has many non-redundant functions; for example, the catalytic activity of the individual JNK isoforms toward different substrates may vary. Indeed, some studies imply different roles of JNK isoforms in particular types of malignant tumors. For example, JNK1, but not JNK2, has been reported to be required for lymphoblast transformation by BCR-ABL [102] and carcinogen-induced hepatocellular carcinoma [103], while some studies implicated JNK2 as the primary isoform involved in glioma cell proliferation [98] and others indicated both JNK1 and JNK2 contribute to maintaining self-renewal and tumorigenicity of glioma CSCs [95,97]. Thus, isoform-specific studies can be vital in assessing the roles of JNK signaling in tumorigenesis as well as in the regulation of CSC phenotypes.

## 6. JNK Signaling in Regulation of Stem Cell Niche Crosstalk

### 6.1. Crosstalk between the Stem Cell Niche and Normal Stem Cells 

Stem cells reside in specific locations in each organ and receive various stimuli from their microenvironments, which maintain their quiescent status, self-renewal, proliferation, or differentiation. This microenvironment supporting stem cells is called the stem cell niche. The stem cell niche is composed of several types of cells, including stromal cells and vascular cells, and the extracellular matrix (ECM) [104]. Growing evidence suggests the importance of this niche for retaining stemness and for decision-making regarding the fate of stem cells, and recent studies showed that JNK signaling could play a role in regulating the crosstalk between stem cells and their niche [105,106]. 

In the adult central nervous system, astrocytes comprise part of the stem cell niche involved in maintaining the stemness of NSCs. Indeed, an in vitro study showed that HMGB1 released from reactive astrocytes stimulates NSC proliferation by binding to the RAGE receptor and activating the JNK signaling pathway [105] (Figure 5A). This astrocyte-NSC communication through the HMGB1/RAGE/JNK signaling pathway has possible beneficial effects in repairing brain injury. 

In embryonic hematopoiesis, JNK signaling also mediates crosstalk between HSCs and their niche. Zhang et al. [106] reported that TGF-β1 promotes HSC emergence by activating JNK signaling in adjacent endothelial cells. Utilizing tgfβ1b-knockout zebrafish, they found that the total Jnk and C-jun protein levels, as well as the expression of Jnk signaling–related genes, including tak1, mapk8, mapk9, and mapk10, were decreased in tgfβ1b-knockout endothelial cells. They also showed that Tgf-β1/C-jun positively controlled the expression level of G6pc3 of the FoxO family, which regulates gluconeogenesis. Furthermore, they demonstrated that glucose metabolism in endothelial cells controlled by the Tgf-β1/c-Jun/G6pc3 cascade plays a critical role in promoting the emergence of HSCs [106] (Figure 5B). Although the JNK signaling pathway may play crucial roles in the functional regulation of fibroblasts [107], endothelial cells [108], and immune cells such as macrophages [109,110] that compose the stem cell niche, there is still little direct evidence showing the importance of JNK signaling in these cells in the context of a supportive niche.

### 6.2. Crosstalk between the Tumor Microenvironment and Cancer Stem Cells

Tumor cells cannot alone cause disease progression and need collaboration with other non-tumor cells existing in the tumor microenvironment (TME) [111]. Similar to the niche for normal stem cells, the TME is known to maintain the stem cell phenotype of CSCs. Furthermore, the TME contributes to resistance to chemotherapy by promoting the stemness of CSCs [112]. For example, cancer-associated fibroblasts secrete various cytokines, including IL-6, which stimulate CSC proliferation and protect them from apoptosis by activating the STAT3 pathway [113,114]. ECM components such as collagen promote CSC self-renewal through the integrin signaling pathway [115] while also forming a thick fiber that serves as a barrier to diffusion of chemotherapeutic drugs [116]. Recent studies [117,118] suggest that JNK signaling also plays a role in forming the CSC niche and regulating crosstalk within the niche. JNK signaling also regulates the CSC niche through inflammatory cells, which play central roles in initiation, proliferation, and metastasis of tumors [119].

Using a myeloid cell-specific Jnk1^−/−^Jnk2^−/−^ mouse model, Han et al. [117] reported that myeloid JNK deficiency reduced inflammatory cytokines (e.g., IL-1β [120], IL-6 [121], and TNF-α [122]) that are known to promote CSC properties in the liver (Figure 5C). Indeed, myeloid JNK deficiency significantly suppressed diethylnitrosamine-induced hepatocellular carcinoma formation in the livers of mice [117]. These data imply that JNK signaling regulates pro-inflammatory cytokine production in inflammatory cells and contributes to maintaining the CSC phenotype, such as tumor initiation and propagation potential. 

Recently, Insua-Rodriguez et al. [118] showed that JNK signaling in breast cancer mediates ECM production and contributes to chemoresistance and metastasis. They demonstrated that constitutive activation of JNK signaling in a human breast cancer cell line promotes significant upregulation of gene expression of ECM proteins such as SPP1 (osteopontin) and TNC (tenascin C). Additionally, JNK signaling promoted stem cell properties, including sphere formation and invasion ability, and treatment of a breast cancer mouse model with paclitaxel-induced SPP1 and TNC expression via JNK signaling. Disruption of SPP1 or TNC expression or pharmacological JNK inhibition by the pan JNK inhibitor CC-401 enhanced the effect of paclitaxel. It reduced lung metastasis in vivo, suggesting that JNK signaling is involved in forming a niche that supports CSC chemoresistance and metastasis [118] (Figure 5D). Overall, JNK signaling affects a variety of cell functions in the TME, such as cytokine and ECM production, and mediates CSC traits. These data suggest that the JNK signaling pathway may be an essential cancer therapeutic target that can modulate the cancer microenvironment. 

## 7. Concluding Remarks

Research on stem cell biology offers new insight into potential applications, such as regenerative medicine and treatments for congenital diseases and malignancies [123]. However, the mechanisms for regulating stem cell traits are still mostly unknown. For example, it is unclear why some types of stem cells tend to be dormant and maintain an undifferentiated status, while others actively proliferate and differentiate. The transcriptional programs for proliferation and differentiation may not be the same among stem cell types, and the signaling pathways which regulate the program also can vary for different types of stem cells. As discussed in this review, the JNK signaling pathway regulates both normal stem cell and CSC properties. The role of JNK signaling in maintaining a self-renewal/undifferentiated state or promoting differentiation is not straightforward, however, and further studies to elucidate the cell context-dependent role of JNK signaling should provide for a better understanding of the characteristics of each stem cell type. 

Significant questions to address are how cells balance the pro-apoptotic and pro-survival/tumorigenic roles of JNK signaling and how differences in the regulation, enzymatic activities, and expression of the JNK isoforms impact their control of stemness. In addition to genetic approaches (e.g., RNA interference, CRISPR-Cas9, or genetically engineered mouse models), there needs to be an emphasis on a better understanding of the differential mechanisms of isoform regulation, the development of isoform-specific JNK inhibitors, as well as the utilization of chemical biology tools that allow for rapid downregulation of specific isoforms using pharmacological approaches [124,125]. Moreover, the crosstalk of the JNK signaling pathway with stemness-related pathways such as WNT and NOTCH, as described in several contexts above, is also largely unexplored and may be vital to unveiling the role of JNK signaling in regulating stemness.

Despite several preclinical studies supporting therapeutic strategies for targeting CSCs, the evidence for clinical efficacy is still limited [126]. Although the classical CSC model assumes an irreversible and straightforward hierarchy of cancer cells, recent studies suggest subclonal genetic diversity among CSCs [127,128]. Also, some studies using newer techniques have found that cells can manifest plasticity between CSC and non-CSC characteristics [129,130]. This genetic diversity and flexibility of CSCs gives rise to complex intratumoral heterogeneity and may make CSC-targeted therapy challenging. One possible solution is targeting the signaling between the CSCs and the TME or CSC niche, which supports the maintenance of the stemness of CSCs. This notion follows from the premise that if the CSC population is dynamic and plastic, CSCs will always re-emerge after targeted ablation if the niche remains intact. JNK signaling, as described above, plays several roles in regulating CSC self-renewal in both CSCs themselves and the CSC niche. Therefore, targeting the JNK signaling pathway has the potential to be an attractive therapeutic strategy in terms of multiple mechanisms, which may eliminate CSCs as a whole by both inhibiting CSC stemness and suppressing the CSC niche. The roles of JNK signaling in normal stem cells and CSCs are still mostly unclear and controversial because of the conflicting oncogene/tumor suppressor results. However, growing evidence suggests a crucial role of JNK signaling in regulating stemness in some types of stem cells or organs/tissues. For example, even though tumor-suppressive roles of JNK signaling are suggested by some reports, including in breast cancer, our group has shown that high JNK activity correlates with poor outcome of TNBC patients and that JNK inhibition reduced CSC proliferation, NOTCH1 expression, and ALDH1 activity in multiple TNBC cell lines as well as in a preclinical mouse model [93]. The next step should be to identify the JNK isoform contributing to CSC and CSC niche regulation in each type of cancer as well as biomarkers to predict the subpopulation of cancer patients who will benefit from JNK-targeted therapy. 

Overall, further studies of JNK signaling and stem cells will be essential for a better understanding of the mechanisms for regulating stemness of normal stem cells and developing novel therapeutic approaches targeting CSCs as promising strategies against malignant tumors.

## Figures and Tables

**Figure 1 ijms-21-02613-f001:**
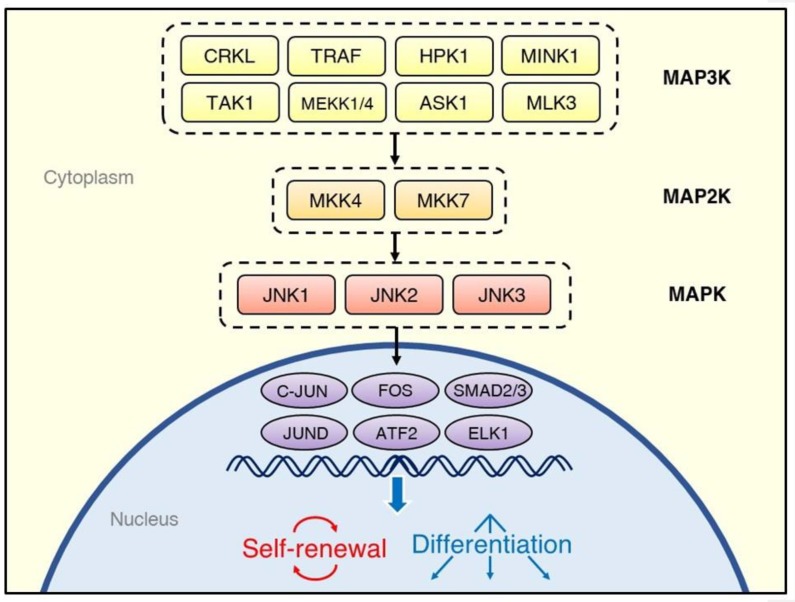
Overview of the regulatory system of the C-JUN N-terminal kinase (JNK) signaling cascade. Various kinds of stimuli, including cytokines, growth factors, and environmental stresses, activate tiers of protein kinases, which phosphorylate JNK isoforms; they, in turn, transduce the signals to transcription factors. The cell fate decisions after activation of the JNK signaling pathway depend on the source of stimulation and type of cells.

**Figure 2 ijms-21-02613-f002:**
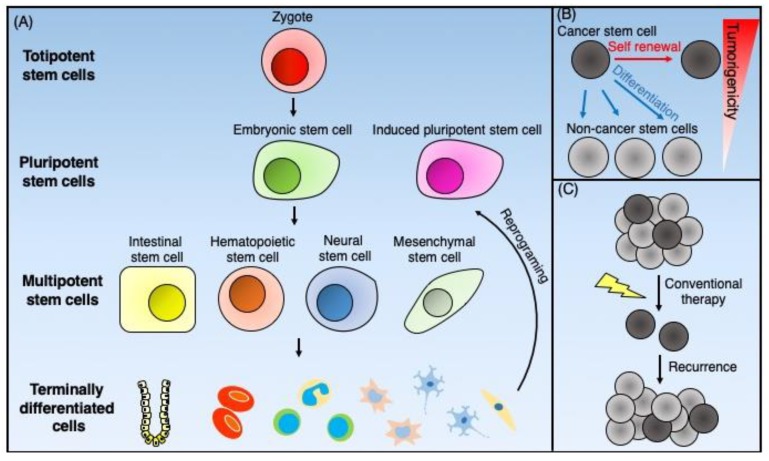
Schematic representation of the normal stem cells and cancer stem cells. (**A**) Totipotent stem cells, such as zygotes, can generate all the types of cells that form total individual organisms. Pluripotent stem cells can give rise to all three germ cell layers. The inner cell mass of blastocysts furnishes embryonic stem cells, and the reprogramming of terminally differentiated cells supplies pluripotent stem cells. Multipotent stem cells can differentiate into organ-specific cell lineages. Adult tissue-specific stem cells such as intestinal stem cells, hematopoietic stem cells, neural stem cells, and mesenchymal stem cells give rise to terminally differentiated cells to maintain tissue homeostasis. (**B**) Cancer stem cells (CSCs) are cancer cells that possess stem cell-like properties. CSCs have both self-renewal ability and differentiation potential into non-CSCs, which have low tumorigenicity. (**C**) CSCs have resistance against conventional chemotherapy and radiotherapy because of multiple mechanisms. As a result of this resistance, CSCs survive such treatments and cause tumor recurrence.

**Figure 3 ijms-21-02613-f003:**
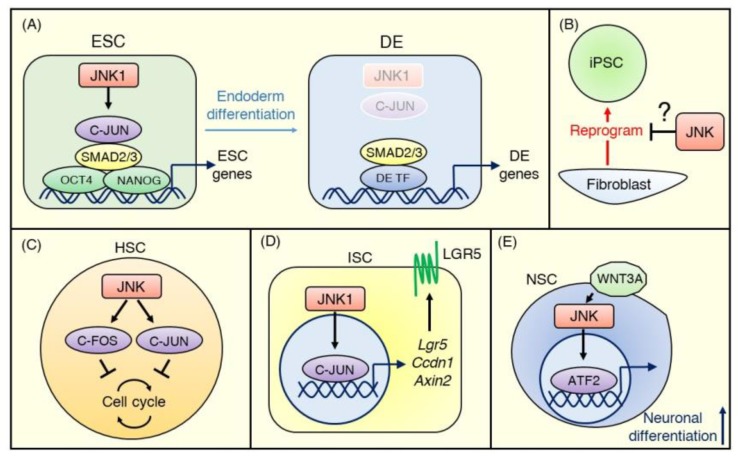
JNK signaling in normal stem cells. (**A**) In embryonic stem cells (ESCs), the JNK pathway inhibits definitive endoderm (DE) differentiation and maintains the undifferentiated status. C-JUN activated by JNK1 binds transcriptional enhancers of ESC genes with OCT4, NANOG, and SMAD2/3 and maintains chromatin accessibility at the ESC stage. During the ESC-DE transition, SMAD2/3 becomes free due to decreased activity of JNK–C-JUN signaling and co-occupies DE enhancers with other DE transcriptional factors. (**B**) The evidence for the role of JNK signaling in iPSCs is minimal. A few conflicting reports suggest that JNK signaling is either suppressed or promoted during the reprogramming of terminally differentiated cells (e.g., fibroblasts) to iPSCs. (**C**) Most hematopoietic stem cells (HSCs) stay quiescent in the absence of stress, and the JNK–C-FOS/C-JUN pathway negatively controls cell cycle progression to maintain a dormant status. (**D**) In ISCs, the JNK signaling pathway modulates WNT signaling strength to regulate intestinal homeostasis. JNK1–C-JUN/AP-1 signaling upregulates WNT target genes, including Lgr5, Ccdn1, and Axin2, to promote proliferation and differentiation of ISCs. (**E**) WNT3A activates the JNK-ATF2 signaling pathway during neuronal differentiation, and ATF2 promotes neuronal gene expression.

**Figure 4 ijms-21-02613-f004:**
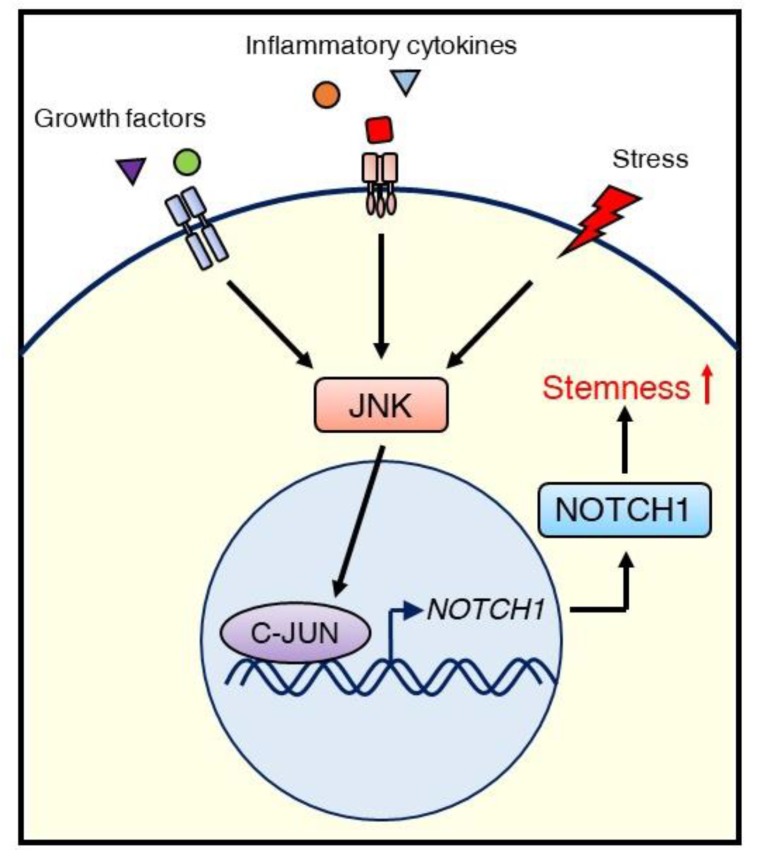
Schematic illustrating the regulation of CSCs via the JNK/C-JUN/NOTCH1 signaling cascade in TNBC. External stimuli, such as growth factors, inflammatory cytokines, or stress, initiate JNK signaling. Activated JNK promotes NOTCH1 expression via C-JUN, leading to stemness (self-renewal), which supports tumor growth.

**Figure 5 ijms-21-02613-f005:**
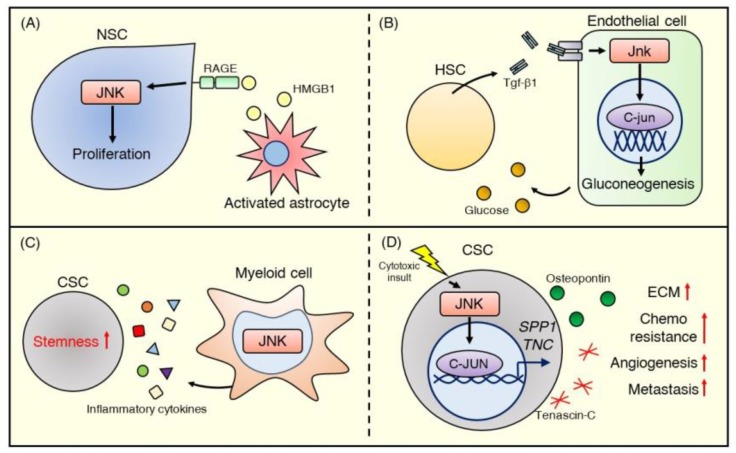
JNK signaling in the stem cell niche. (**A**) HMGB1 released from activated astrocytes stimulates the RAGE-JNK pathway and promotes the proliferation of NSCs. (**B**) Embryonic HSCs stimulate endothelial cells with Tgf-β1 to encourage glucose production and HSC emergence via activating the Jnk–C-jun signaling pathway. (**C**) JNK signaling contributes to myeloid cell function in the CSC niche. JNK-deficient myeloid cells in the liver exhibit reduced secretion of inflammatory cytokines (e.g., IL-1β, IL-6, and TNF-α), which promote CSC stemness. (**D**) JNK signaling in breast cancer mediates ECM production. The JNK–C-JUN pathway, stimulated by cytotoxic stress, upregulates gene expression in the ECM, wound healing, and stem cells and promotes the production of matricellular proteins such as osteopontin and tenascin C, encoded by SPP1 and TNC, respectively. These proteins activate angiogenesis and further ECM production in the CSC niche and subsequently promote metastasis and chemoresistance.

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
