# Peer review of "JNK Signaling in Stem Cell Self-Renewal and Differentiation"

_ijms, 2020, doi:10.3390/ijms21072613_

Round 1

Reviewer 1 Report

Perhaps I was not clear in my previous comment about  mesenchymal stem cells.

I wrote:

A minor comment on the term mesenchymal stem cells. Authors have to report a detailed and proper definition for mesenchymal stromal cells (MSCs). Stromal cells are heterogeneous and contain several populations, including stem cells. The term MSCs should be used for mesenchymal stromal cells rather than mesenchymal stem cells since the cells they isolated do not contain a pure population of stem cells. Indeed, the authors should better explain that the isolation of MSCs according to current criteria produces heterogeneous, non-clonal cultures of stromal cells containing stem cells with different multipotential properties, committed progenitors, and differentiated cells. To help in proper definitions please see: Cell Transplant. 2016;25(5):829-48. doi: 0.3727/096368915X689622. PubMed PMID: 26423725.

The author reply did not catch my observation. They wrote:

 We apologize for our wrong use of the term MSCs. In the revised manuscript, we have corrected the sentence to clarify that mesenchymal stem cells, but not mesenchymal stromal cells, are discussed as shown below. Page 4, line 105-107: Adult tissue-specific stem cells are considered multipotent. These include hematopoietic stem cells (HSCs), intestinal stem cells (ISCs), neural stem cells (NSCs), and mesenchymal stem cells.

The authors have to understand that the problem is not  the use of MSCs for mesenchymal stem cells. The problem is the use of the term mesenchymal stem cells. I mean that all studies claiming the use of mesenchymal stem  cells    are not correct since people do not use pure stem cell population. Indeed cells that are collected from bone marrow or adipose tissue are mesenchymal stromal cells, which contain a subpopulation of stem cells. Please read again what i wrote before and correct the manuscript using the term mesenchymal stromal cells. Reading the  paper I cited may help you in understanding this issue.

Author Response

We apologize for our misunderstanding. According to the suggestion, we have defined and discussed about mesenchymal stromal cells with incorporating the previous works including one provided by the reviewer.

Page 4, line 105-115: Adult tissue-specific stem cells are considered multipotent. These include hematopoietic stem cells (HSCs), intestinal stem cells (ISCs), and neural stem cells (NSCs). These cells are present in the respective tissues, where they maintain tissue regeneration and participate in wound repair [51]. Multipotent mesenchymal stromal cells (MSCs) are the fibroblast-like plastic-adherent cells derived from bone marrow and other tissues, which contain a subpopulation of stem cells of mesenchymal lineages [52]. According to the criteria proposed by the International Society for Cellular Therapy, cultured MSCs must express CD105, CD73 and CD90, and lack expression of CD45, CD34, CD14 or CD11b, CD79a or CD19 and HLA-DR surface molecules and possess the ability to differentiate into osteocytes, chondrocytes, and adipocytes in vitro [53]. It should be noted that current standard culture conditions for MSCs do not isolate homogenous stem cell populations [54].

We also have included the additional sources to address the reviewer comment:

Horwitz EM et al. Clarification of the nomenclature for MSC: The International Society for Cellular Therapy position statement. Cytotherapy 2005, 7, (5), 393-395.

Dominici M et al. Minimal criteria for defining multipotent mesenchymal stromal cells. The International Society for Cellular Therapy position statement. Cytotherapy 2006, 8, (4), 315-317.

Squillaro T et al. Clinical Trials With Mesenchymal Stem Cells: An Update. Cell Transplant. 2016, 25, (5), 829-848.

Reviewer 2 Report

Authors fully answered to reviewers' comments.

Author Response

We are grateful that the reviewer has accepted our revised manuscript.

This manuscript is a resubmission of an earlier submission. The following is a list of the peer review reports and author responses from that submission.

Round 1

Reviewer 1 Report

The concise review on "JNK signaling in stem cell self renewal and differentiation" addresses a very important and "hot" issue in stem cells research, i.e. the role of JNK signalling in self renewal and differentiation.

The manuscipt is well-organized, as first authors make an overview of JNKs and of all the MAPK pathway, than they carefully introduce the different types of stem cells. I suggest to complete this part of the review by adding a figure with a graphical representation of stem cells.

Than, authors focus their attention on adult stem cells, but it should be interesting for readers to add also a brief description of JNKs role in embryonic stem cells.

Regarding the role of JNKs in neural stem cells, the description of JNKs should also include the STAT role.

Reviewer 2 Report

In the present paper the authors presented a concise review on JNK signaling in stem cell self-renewal and differentiation.

The paper is of interest and it describes clearly the role of JNK in stem cell biology. I have only a few comments.

Premature senescence occurs when normal human somatic cells irreversibly lose their self-renewal capability after completing a limited number of cell divisions. There are several papers showing that JNK may play a role in preventing premature senescence in cancer stem cells (Oncogene volume 29, pages561–575 2010) or alternatively in promoting survival of senescent cells (EMBO J. 2017 Aug 1; 36(15): 2280–2295). A comment on this issue would further improve the manuscript.

A minor comment on the term mesenchymal stem cells. Authors have to report a detailed and proper definition for mesenchymal stromal cells (MSCs). Stromal cells are heterogeneous and contain several populations, including stem cells. The term MSCs should be used for mesenchymal stromal cells rather than mesenchymal stem cells since the cells they isolated do not contain a pure population of stem cells. Indeed, the authors should better explain that the isolation of MSCs according to current criteria produces heterogeneous, non-clonal cultures of stromal cells containing stem cells with different multipotential properties, committed progenitors, and differentiated cells. To help in proper definitions please see: Cell Transplant. 2016;25(5):829-48. doi: 0.3727/096368915X689622. PubMed PMID: 26423725.